

# Development of a relapse-related RiskScore model to predict the drug sensitivity and prognosis for patients with ovarian cancer

Zhixin Jin[1,*], Xuegu Wang[2,*], Xiang Li[3], Shasha Yang[1], Biao Ding[2], Jiaojiao Fei[1], Xiaojing Wang[1] and Chengli Dou[3]

[1] Anhui Key Laboratory of Respiratory Tumors and Infectious Diseases, The First Affiliated Hospital of Bengbu Medical University, Bengbu, China
[2] Department of Obstetrics and Gynecology (Center for Reproductive Medicine), The First Affiliated Hospital of Bengbu Medical University, Bengbu, China
[3] Molecular Diagnostic Center, The First Affiliated Hospital of Bengbu Medical University, Bengbu, China
[*] These authors contributed equally to this work.

Corresponding authors
Xiaojing Wang,
wangxiaojing8888@163.com
Chengli Dou, hzaudou2018@163.com

## ABSTRACT

**Background**. Ovarian cancer (OC) is a highly aggressive malignancy in the reproductive system of women, with a high recurrence rate. The present research was designed to establish a relapse-based RiskScore model to assess the drug sensitivity and prognosis for patients with OC.

**Methods**. Gene Expression Omnibus (GEO) and The Cancer Genome Atlas (TCGA) databases were accessed to obtain relevant sample data. The single-cell atlas of primary and relapse OC was characterized using the "Seurat" package. Differentially expressed genes (DEGs) between primary and relapse samples were identified by FindMarkers function. Subsequently, univariate Cox, least absolute shrinkage and selection operator (LASSO) and stepwise regression analysis were employed to determine independent prognostic genes related to relapse in OC to establish a RiskScore model. Applying "timeROC" package, the predictive performance of RiskScore model was assessed. Drug sensitivity of different risk groups was evaluated using "pRRophetic" package. The effects of relapse-related prognostic genes on OC cells were detected with *in vitro* assays.

**Results**. The single-cell atlas revealed that compared to primary OC, fibroblasts were reduced but epithelial cells were increased in relapse OC. Five prognostic genes (*LDHA*, *NOP58*, *NMU*, *KRT19*, and *RPS23*) independently linked to relapse in OC were identified to construct a RiskScore model, which showed high robustness in the prognostic prediction for OC patients. High-risk group tended to have worse outcomes in terms of different clinical features than the low-risk group. Further, six drugs (Vinorelbine, GW-2580, S-Trityl-L-cysteine, BI-2536, CP466722, NSC-87877) were found to be correlated with the RiskScore. While the high-risk group had higher IC$_{50}$ values to these drugs, the low-risk group was more sensitive to the six drugs. In addition, *KRT19* silencing markedly inhibited the invasion and migration of OC cells.

**Conclusion**. This study established a relapse-related RiskScore model based on five prognostic genes (*LDHA*, *NOP58*, *NMU*, *KRT19*, and *RPS23*), offering novel insights into the recurrence mechanisms in OC and contributing to the development of individualized treatment strategies.

# INTRODUCTION

Ovarian cancer (OC) is a fatal female reproductive system malignancy that shows a 5-year overall survival (OS) rate lower than 50% (*Ma, Shao & Zhu, 2024*; *Jiang et al., 2024*). Global Cancer Observatory reports showed that OC accounted for about 313,959 new cases and 207,252 deaths globally in 2020 (*Salima et al., 2022*; *Ma et al., 2023*). Moreover, non-specific symptoms and absence of reliable methods for screening have caused a late-stage diagnosis (FIGO III or IV) in about 75% of OC patients, who often have extensive intra-abdominal metastasis (*Cheung et al., 2020*). OC is considered as a particularly challenging cancer due to its genetic and non-genetic risk factors (*Hua et al., 2023*). At present, the standard-of-care strategy for primary OC is cytoreductive surgery combined with platinum/paclitaxel adjuvant chemotherapy (*Wang et al., 2024a*). Recently, some new regimens such as intraperitoneal injection and targeted therapies (such as poly ADP-ribose polymerase inhibitors) have emerged as the first-line treatments for OC (*Emmings et al., 2019*). However, most patients with advanced OC still have unfavorable survival, pointing to the need to develop novel molecular indicators for OC.

The progression of OC can be inhibited if treated properly but subsequent chemoresistance and recurrence remain difficult to be overcome (*Chesnokov, Yadav & Chefetz, 2022*), which is also a major contributor to a worse prognosis (*Boylan et al., 2020*). Recurrence occurs to around 25% of OC patients in early-stage within 6 months of initial treatment (*Zhao et al., 2021b*), and around 70% of OC patients in advanced-stage relapse within 3 years (*Johnston et al., 2023*). Most of OC patients ultimately develop platinum-resistance (*Howard et al., 2020*), which will notably reduce the therapeutic effects. Hence, there is an increasing need to explore the molecular mechanism of OC recurrence and develop novel sensitive drugs for OC patients. Recent single-cell RNA-sequencing (scRNA-seq) analysis has provided novel understanding of OC development (*Zhao et al., 2021a*; *Kodous, Balaiah & Ramanathan, 2023*). For instance, previous study applied scRNA-seq and bulk expression dataset to establish a recurrence-related 13-gene risk model for assessing OC prognosis (*Zhang et al., 2022*). However, the etiopathology associated with relapse in OC is incomprehensively understood.

The present study downloaded OC samples from The Cancer Genome Atlas (TCGA) and Gene Expression Omnibus (GEO) databases. The single-cell atlas of primary and relapse OC was then characterized by scRNA-seq analysis. Thereafter, prognostic genes related to relapse in OC were identified through univariate Cox, lease absolute shrinkage and selection operator (LASSO) Cox and stepwise regression analysis. Afterwards, a RiskScore model was constructed and verified under different clinical features. In addition, we assessed the differences of enriched pathways and drug sensitivity between the risk groups. *In vitro* experiments were also utilized to evaluate the effects of the relapse-related prognostic genes

on OC cells. The current findings were expected to provide novel understanding to the improvement of personalized treatment of OC and the prognostic prediction.

## MATERIAL AND METHODS

### Acquisition and preprocessing of data

The scRNA-seq dataset GSE130000 (https://www.ncbi.nlm.nih.gov/geo/query/acc.cgi?acc=GSE130000), which contained four primary and two relapse OC samples, was obtained from the GEO database.

The TCGA database (https://portal.gdc.cancer.gov) was accessed to collect the clinical data and gene expression data of OC patients. Then, the FPKM value of RNA-seq data was converted into transcripts per million (TPM) and then log2-tranformed. A total of 177 OC samples with disease-free interval (DFI) collected from the TCGA-OC cohort served as the training set.

The GSE63885 dataset (GPL570 platform, https://www.ncbi.nlm.nih.gov/geo/query/acc.cgi?acc=GSE63885) containing the gene expression data and clinical information of OC patients was downloaded from the GEO database as well. A total of 101 OC samples with complete DFI information in the GSE63885 dataset served as the validation set.

### Analysis of scRNA-seq data

The single-cell atlas of primary and relapse OC was characterized based on the scRNA-seq data in GSE130000 dataset applying the "Seurat" R package (Zulibiya et al., 2023). The Read10X function in Seurat was employed to read the scRNA-seq data of each OC sample and retain cells with gene number between 200 and 1,500. Then, after data normalization using SCTransform function, dimensionality reduction was achieved applying principal component analysis (PCA), followed by removing batch effects among different OC samples with the "harmony" R package (Mu et al., 2024; Chen et al., 2025). Next, uniform manifold approximation and projection (UMAP) analysis was carried out using the RunUMAP function for dimensionality reduction. The FindNeighbors and FindClusters functions (parameters: dims = 1:20 and resolution = 0.1) were employed to cluster the cells. Furthermore, cell types were annotated utilizing the marker genes from the CellMarker2.0 database.

### Differentially expressed genes analysis

The differentially expressed genes (DEGs) between primary and relapse OC samples in GSE130000 dataset were identified by the FindMarkers function. Next, Gene Ontology (GO) enrichment analysis in biological process (BP) term was conducted on the DEGs applying the gseGO function in "clusterProlifer" R package (Ding et al., 2022b).

### Development and validation of RiskScore model

Relapse-related prognostic genes for OC were screened from the DEGs using univariate Cox proportional hazard regression analysis ($p < 0.05$) and further refined by LASSO Cox regression analysis in "glmnet" R package (Li et al., 2024). Then, the prognostic genes independently related to relapse in OC were determined by stepwise regression analysis

and used to create a RiskScore model as follow (*Yu, Zhao & Yu, 2024*):

$$RiskScore = \sum \beta i^*ExPi$$

$\beta i$ refers to the coefficient of a gene in Cox regression model, ExPi denotes the gene expression value.

OC patients were split by the median value of RiskScore into low- and high-risk groups. DFI and OS between different risk groups were compared using Kaplan–Meier (K–M) curve. The receiver operating characteristic (ROC) curve was plotted by the "timeROC" R package (*Gao, Zhang & Tian, 2024*) to test the predictive accuracy of the RiskScore model. Additionally, the accuracy and robustness of RiskScore was verified in GSE63885 dataset.

### Gene set enrichment analysis

Differences of the enriched pathways between the two risk groups were compared by Gene Set Enrichment Analysis (GSEA) (*Jia et al., 2024*) under the screening threshold of false discovery rate (FDR) < 0.05. The HALLMARK pathway of "h.all.v7.5.1.symbols.gmt" downloaded from the MSigDB database (https://www.gsea-msigdb.org/gsea/msigdb/) served as the background gene set. Single-sample GSEA (ssGSEA) with the "GSVA" R package (*Yi et al., 2023*) was employed to compute the HALLMARK enrichment scores for different risk groups.

### Analysis of drug sensitivity

Applying the "pRRophetic" R package, the half-maximal inhibitory concentration ($IC_{50}$) of chemotherapy agents in different risk groups in TCGA-OC cohort was assessed (*Geeleher, Cox & Huang, 2014*). Moreover, the correlation between the $IC_{50}$ and RiskScore was examined to screen drugs with $p < 0.05$ and |cor|>0.3.

### Cell line culture and transfection

To conduct *in vitro* assays for validation, human normal ovarian epithelial cell line (IOSE-80, XY-H366) and OC cell line (SK-OV-3, XY-H005) were ordered from the Shanghai Xinyu Biotechnology Co., Ltd. Specifically, IOSE-80 cells were cultivated in the RPMI-1640 medium (XY-HM366) with 10% fetal bovine serum (FBS) and penicillin/streptomycin (P/S). SK-OV-3 cells were cultured in the DMEM medium (XY-HM005) with 10% FBS and P/S. All the cell lines were placed in the incubators with 5% $CO_2$ at 37 °C for subsequent experiments.

Shanghai Sangon Biological Engineering Co., Ltd provided the siRNA of *KRT19* (si-*KRT19*: 5′-AAGATCACTACAACAATTTGTCT-3′) and negative control (si-NC). For silencing *KRT19*, Lipofectamine 2000 Reagent (Invitrogen, Carlsbad, CA, USA) was utilized to transfect the SK-OV-3 cell lines with siRNA.

### Quantitative real-time PCR

The relative mRNA expressions of the key prognostic genes in IOSE-80 and SK-OV-3 cells were detected by real time quantitative polymerase chain reaction (RT-qPCR) test. Firstly, Trizol (abs9331-1kit, Absin, Shanghai, China) was employed for isolating total RNA, which was subsequently reverse-transcribed into cDNA employing First Strand cDNA Synthesis

**Table 1 Primer sequences utilized in RT-qPCR assay.**

| Gene | Primers (5′–3′) |
|---|---|
| NMU | Forward: GTCAAATGTTGTGTCGTCAGTTG |
| | Reverse: CCATTCCGTGGCCTGAATAAA |
| RPS23 | Forward: GGTGCTTCTCATGCAAAAGGA |
| | Reverse: GCAACCGTCATTGGGTACAAA |
| KRT19 | Forward: TGAGTGACATGCGAAGCCAAT |
| | Reverse: CTCCCGGTTCAATTCTTCAGTC |
| LDHA | Forward: ATGGCAACTCTAAAGGATCAGC |
| | Reverse: CCAACCCCAACAACTGTAATCT |
| NOP58 | Forward: ACAGAAAGTTGGCGATAGGAAG |
| | Reverse: AGCTGGGTTCGATATTCAGAGA |
| GAPDH | Forward: CTGGGCTACACTGAGCACC |
| | Reverse: AAGTGGTCGTTGAGGGCAATG |

Kit (abs601510, Absin, Shanghai, China). Thereafter, RT-qPCR was conducted with the SYBR Green I (abs47043115, Absin, Shanghai, China) (*Zhang et al., 2023a*). See Table 1 for primer sequences of this study. *GAPDH* was an internal control for computing the gene expression using $2^{-\Delta\Delta CT}$ method (*Nomiri et al., 2022*). All the samples were analyzed in triplicate.

## Cell invasion and migration assays

Transwell assay was employed to detect the invasion of OC cells with *KRT19* silencing. Specifically, Matrigel was firstly pre-coated on the Transwell chamber (pore: 8.0 μm, Corning, Inc., Corning, NY, USA). Then, the transfected OC cells ($5 \times 10^5$ cells/well) were inoculated into the upper chamber, while the lower chamber contained 600 μL DMEM with 10% FBS and P/S. After incubation for 48 hours (h), the OC cells in the lower Transwell chamber were fixed by 4% paraformaldehyde, dyed by 0.1% crystal violet. Wound healing assay was conducted to evaluate the migratory ability of *KRT19*-silenced OC cells. Briefly, the OC cells ($2 \times 10^4$ cells/well) were cultured in 6-well plate to form a full confluent monolayer. Hereafter, the OC cell monolayer was wounded by sterile tips and continued to be cultured for another 48 h. Finally, an Olympus IX71 inverted microscope was applied to capture representative photographs, and invaded cell number and wound closure (%) of OC cells were quantified and measured by ImageJ (x64) 1.8.0 software (*Yu et al., 2023*; *Zhang et al., 2024*).

## Statistical analysis

R software (version3.6.0) and GraphPad Prism (version 8.0) were employed in all statistical analyses. Wilcoxon rank-sum test was utilized to compare differences between two continuous variables, whereas survival differences across different risk groups were compared with log-rank test. Spearman method was employed in correlation analysis. A $p < 0.05$ denoted statistical significance, and the data were shown in mean ± standard deviation. Two-way analysis of variance and unpaired *t* test were applied to compare the experimental data.

## RESULTS

### Single-cell atlas of primary and relapse OC was characterized

The scRNA-seq analysis was utilized to delineate the single-cell atlas of primary and relapse OC in GSE130000 dataset. After cell filtering, data normalization, dimensionality reduction, and clustering analysis, a sum of 23,079 cells were divided and annotated into four cell types (Fig. 1A), including fibroblasts (*COL1A1*, *COL3A1*, *DCN*), endothelial cells (*SPARCL1*, *VWF*), myeloid cells (*SPP1*, *CD74*), and epithelial cells (*WFDC2*, *CD24*, *KRT18*, *KRT19*, *EPCAM*) (Fig. 1B). Compared with primary OC samples, the proportion of fibroblasts was markedly reduced and that of epithelial cells was notably increased in relapse OC samples (Figs. 1C and 1D). Furthermore, GO enrichment analysis showed that the DEGs between primary and relapse OC samples were mainly enriched in cytoplasmic translation, G protein-coupled receptor signaling pathway, cellular amide metabolic process, peptide metabolic process, cell–cell adhesion, response to chemical, *etc.* in GO-BP term (Fig. 1E), indicating that these biological processes may play crucial roles in the recurrence of OC.

### Identification of key relapse-related genes

Initial screening using univariate Cox proportional hazard regression analysis selected 27 prognostic genes linked to relapse OC ($p < 0.05$), including 16 risk genes with hazard ratio (HR) > 1 and 11 protective genes with HR < 1 (Fig. 2A). Subsequently, the candidate genes in the model were refined by LASSO Cox regression analysis (Figs. 2B and 2C). Applying stepwise regression analysis, five independent genes related to relapse in OC were identified, including 2 protective genes with HR < 1 (*LDHA*, *NOP58*) and 3 risk genes with HR > 1 (*NMU*, *KRT19*, *RPS23*) (Figs. 2D and 2E). These genes were employed to formulate the RiskScore model: Riskscore = 0.29 * NMU + 0.39 * RPS23 + 0.33 * KRT19−0.48 * LDHA−0.41 * NOP58.

### Construction of a relapse-related RiskScore model and prediction of prognosis in OC

All the patients in the TCGA-OC cohort were separated by the median value of RiskScore into high- and low-risk groups. The area under ROC curve (AUC) was calculated to test the prediction performance of the RiskScore model. It was observed that in TCGA-OC cohort, the RiskScore model had 1-year AUC of 0.68, 2-year AUC of 0.71, 3-year AUC of 0.74, 4-year AUC of 0.75, and 5-year AUC of 0.82, showing a high performance in predicting OC patients' prognosis (Fig. 3A). K-M curves displayed that high-risk group had lower DFI and OS than low-risk group in TCGA-OC cohort (Figs. 3B and 3C), indicating a worse prognosis in high-risk OC group. Additionally, the RiskScore model reached 1-year AUC of 0.61, 2-year AUC of 0.74, 3-year AUC of 0.69, and 4-year AUC of 0.74 in GSE63885 dataset (Fig. 3D). Similarly, compared to low-risk group, high-risk group had a shorter DFI and worse outcome in GSE63885 dataset (Fig. 3E). These results validated the accuracy and robustness of RiskScore model in the prognostic evaluation in OC.

### Validation of the expressions of the five key prognostic genes in OC

The expressions of the five independent prognostic genes were compared in different risk groups. In both TCGA-OC cohort and GSE63885 dataset, *NMU*, *RPS23*, and *KRT19* were

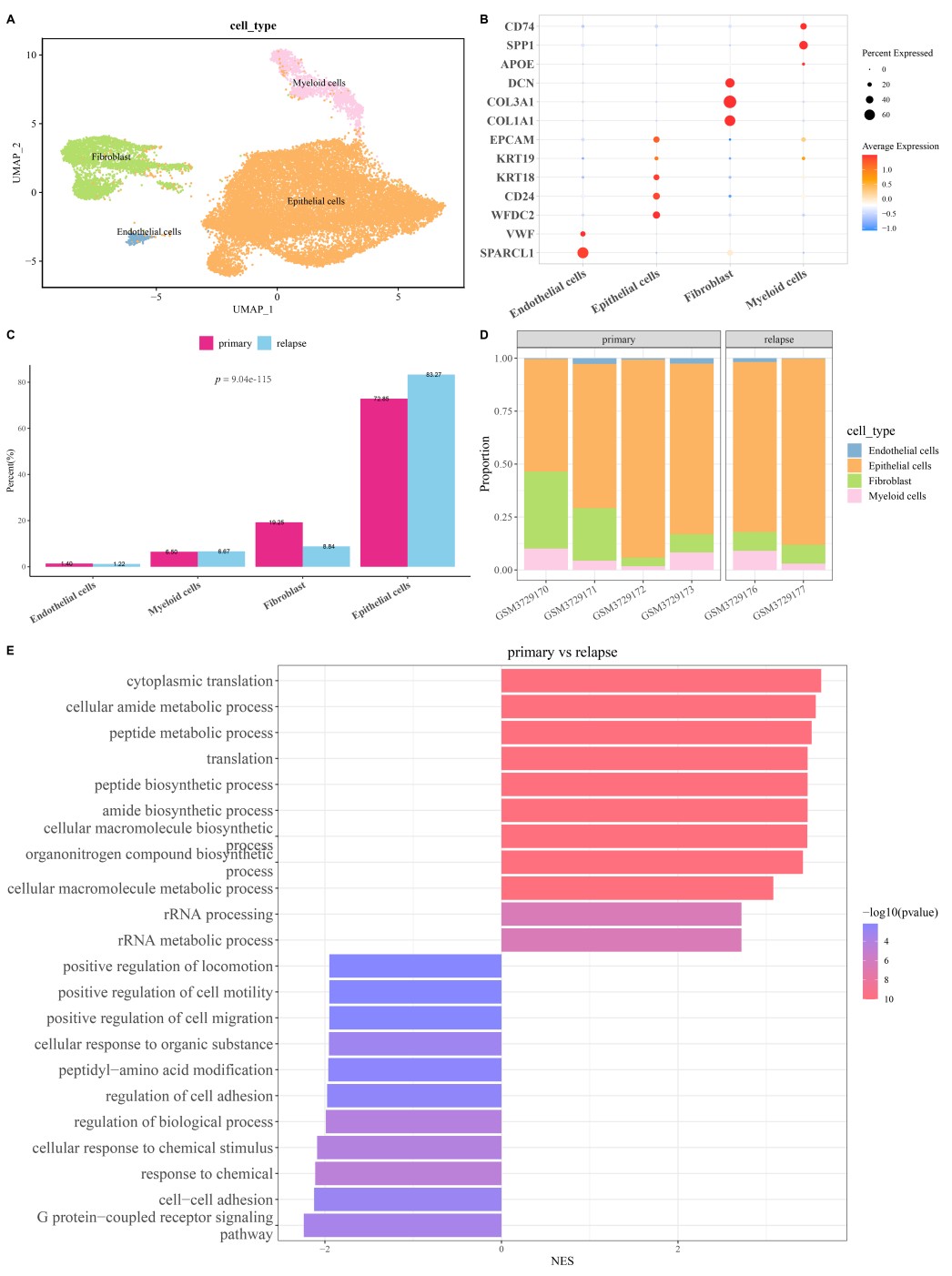

**Figure 1 Single-cell atlas of primary and relapse ovarian cancer (OC).** (A) OC cell types after annotation shown in UMAP plot; (B) Marker genes of each cell type; (C–D) The proportion of each cell type in primary and relapse OC samples; (E) Differences in biological processes (BP) between primary and relapse OC samples.

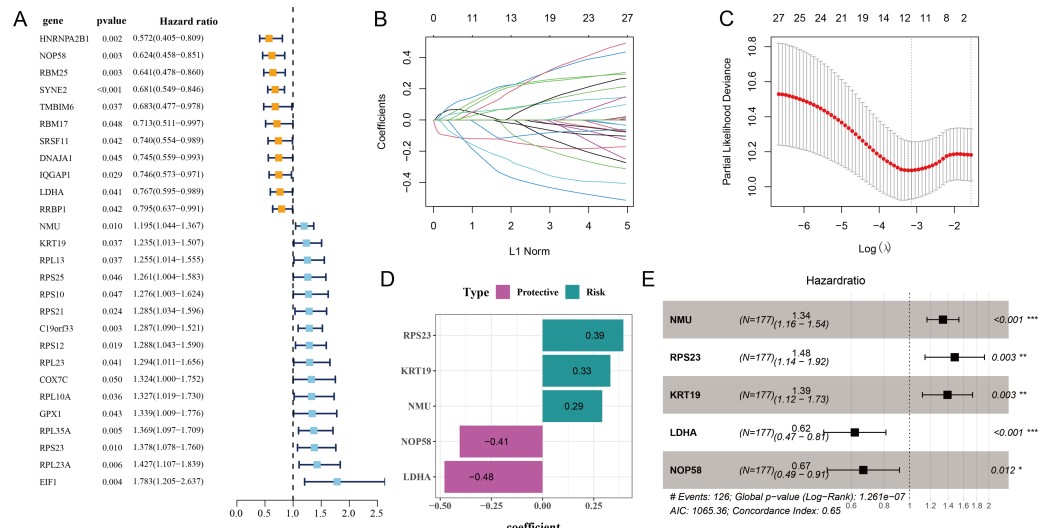

**Figure 2** **Screening of independent prognostic genes related to relapse in OC.** (A) Univariate Cox regression analysis of relapse-related prognosis genes; (B–C) LASSO Cox regression analysis to reduce gene number; (D) Independent prognostic genes related to relapse in OC and their coefficients; (E) Hazard ratio and 95% confidence interval of independent prognostic genes; * represents $p < 0.05$; ** represents $p < 0.01$; *** represents $p < 0.001$.

all high-expressed but *LDHA* and *NOP58* were low-expressed in high-risk group compared with low-risk group (Figs. 4A and 4B). Moreover, the expression pattern of the five genes in single-cell dataset GSE130000 was further analyzed. *RPS23* was high-expressed in the Epithelial cells, Fibroblasts and Myeloid cells of primary OC samples, whereas *NMU*, *KRT19*, *LDHA*, and *NOP58* were mainly expressed in relapse OC samples (Figs. 4C and 4D).

## Evaluation of the predictive performance of the RiskScore model in different clinical characteristics of OC patients

The prognostic differences between low- and high-risk OC groups with different clinical characteristics were compared. It was found that high-risk group had shorter DFI and worse prognosis with different Age (Figs. 5A and 5B), Stage (Figs. 5C and 5D), and Grade (Figs. 5E and 5F). These outcomes further demonstrated that the present RiskScore model was independent and reliable in the prognostic prediction for OC patients with different clinical features.

## GSEA and drug sensitivity analysis in different risk groups

Compared to low-risk group, GSEA revealed that high-risk group had notably activated coagulation pathway and observably suppressed pathways of G2M checkpoint, E2F targets, mitotic spindle, Wnt beta catenin signaling, PI3K AKT mTOR signaling, interferon alpha response, and interferon gamma response (Fig. 6A). In addition, we screened six chemotherapy drugs closely linked to RiskScore (|cor| > 0.3), namely, Vinorelbine, GW-2580, S-Trityl-L-cysteine, BI-2536, CP466722, and NSC-87877 (Fig. 6B). The IC$_{50}$ values

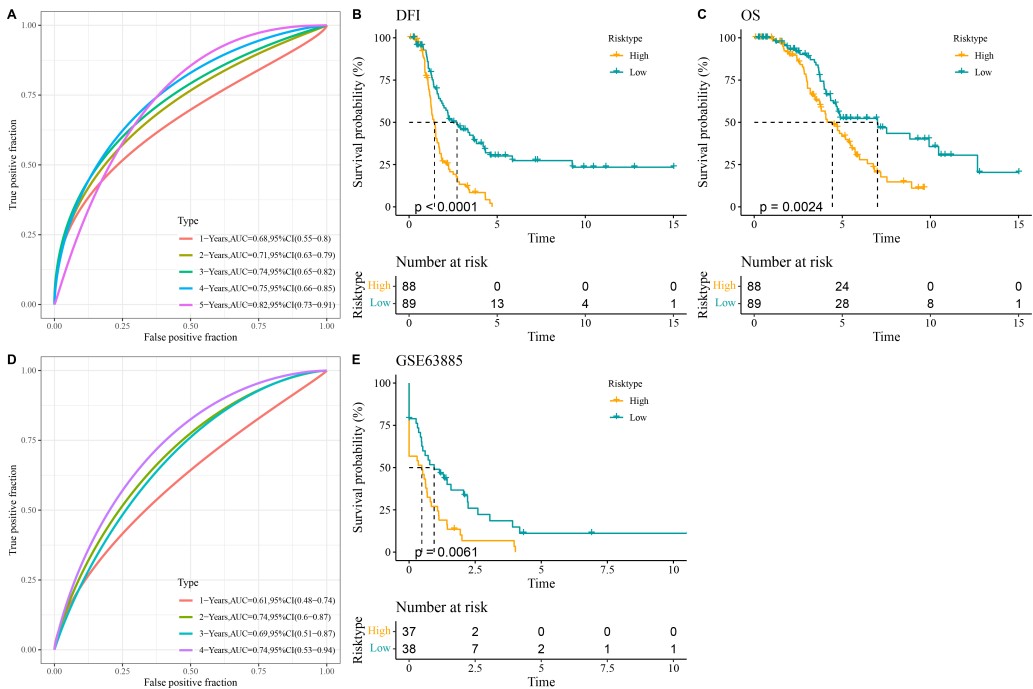

**Figure 3** **Construction and validation of RiskScore model.** (A) The performance of RiskScore model in TCGA-OC cohort reflected by receiver operating characteristic (ROC) curve; (B) Disease-free interval (DFI) in Kaplan–Meier (K–M) curve analysis for different risk groups in TCGA-OC cohort; (C) Overall survival (OS) in K–M curve analysis for different risk groups in TCGA-OC cohort; (D) ROC curve of RiskScore model in GSE63885 dataset; (E) DFI in K–M curve analysis for different risk groups in GSE63885 dataset.

of the six chemotherapy drugs were all markedly higher in high-risk group in comparison to low-risk group (Fig. 6C), which indicated that OC patients in low-risk group had higher sensitivity to these drugs. These findings could provide novel insights for the drug selection in the personalized treatment of OC.

### *In vitro* cellular experiments to validate the expression and potential function of the screened key genes

RT-qPCR assay showed that compared to human normal ovarian epithelial cells IOSE-80, three genes related to relapse (*KRT19*, *LDHA*, *NOP58*) were all significantly upregulated in OC cells SK-OV-3 (Fig. 7A). As *KRT19* expression was most significantly upregulated in OC cells and its pro-carcinogenic effects on tumors have been previously reported (*Sun et al., 2023b*; *Shi et al., 2024*), *KRT19* was therefore selected as a representative gene in this study and *in vitro* experiments were carried out to validate its potential function in the metastatic process of OC cells. Hereafter, *KRT19* was used for silencing to evaluate its impacts on OC cell invasion and migration *via* Transwell and Wound healing assays. The number of invaded cells and wound closure rate of SK-OV-3 were markedly reduced after silencing *KRT19*

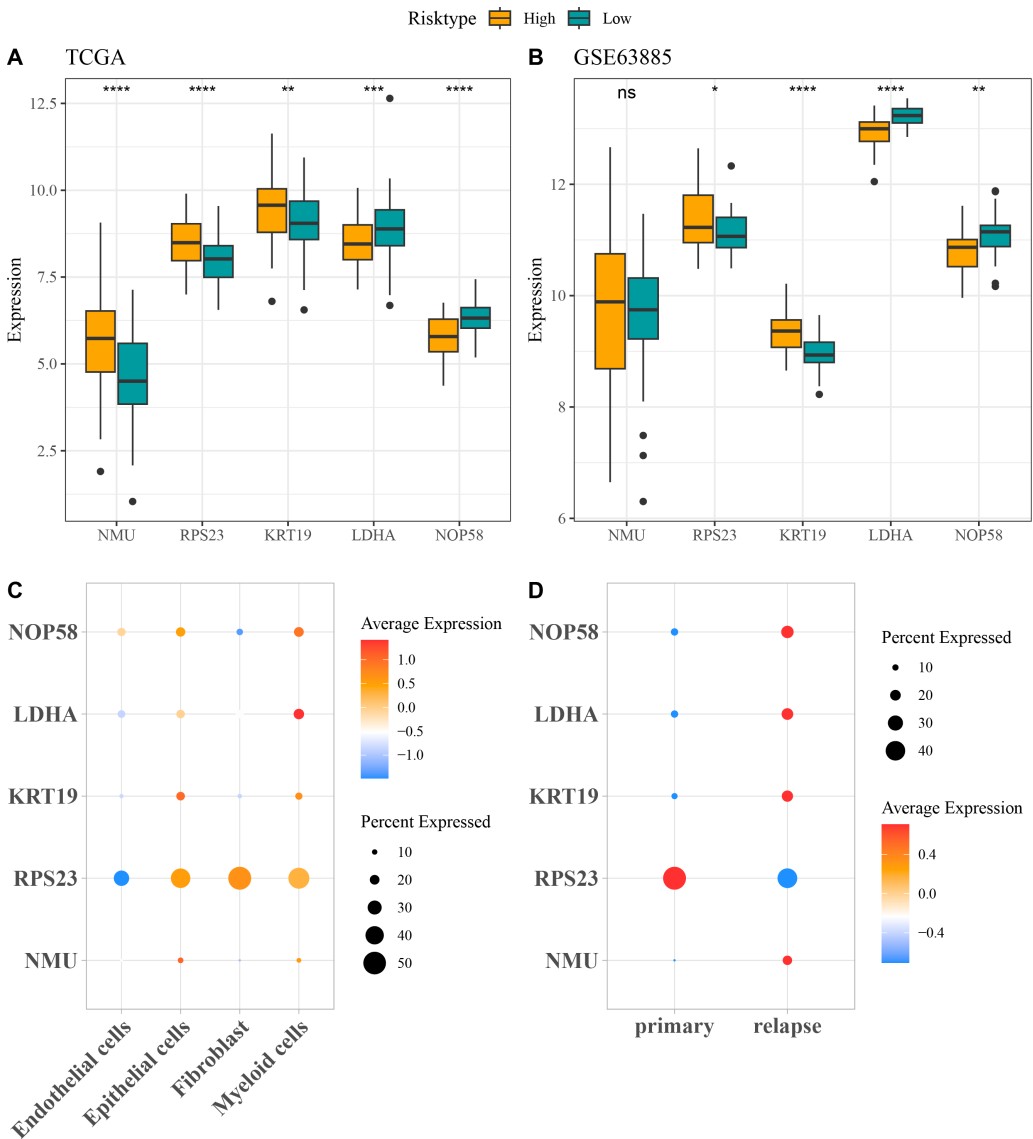

**Figure 4** **Expressions of the independent prognostic genes associated with relapse in OC.** (A) The expressions of independent prognosis genes in different risk groups in TCGA-OC cohort; (B) The expressions of independent prognostic genes in different risk groups in GSE63885 dataset; ns means not significant; **** means $p < 0.0001$; *** means $p < 0.001$; ** means $p < 0.01$; * means $p < 0.05$; (C) Expression pattern of independent prognosis genes in different cell types in GSE130000 dataset; (D) Expression pattern of independent prognosis genes in primary and relapse OC samples in GSE130000 dataset.

(Figs. 7B–7E). These findings suggested that *KRT19* might play an imperative role in the OC developed.

## DISCUSSION

OC is a prevalent malignant tumor in gynecology with high metastasis and recurrence rates and unfavorable survival (*Song & Qu, 2022*; *Zheng, Li & Zhan, 2022*). Hence, it is of great

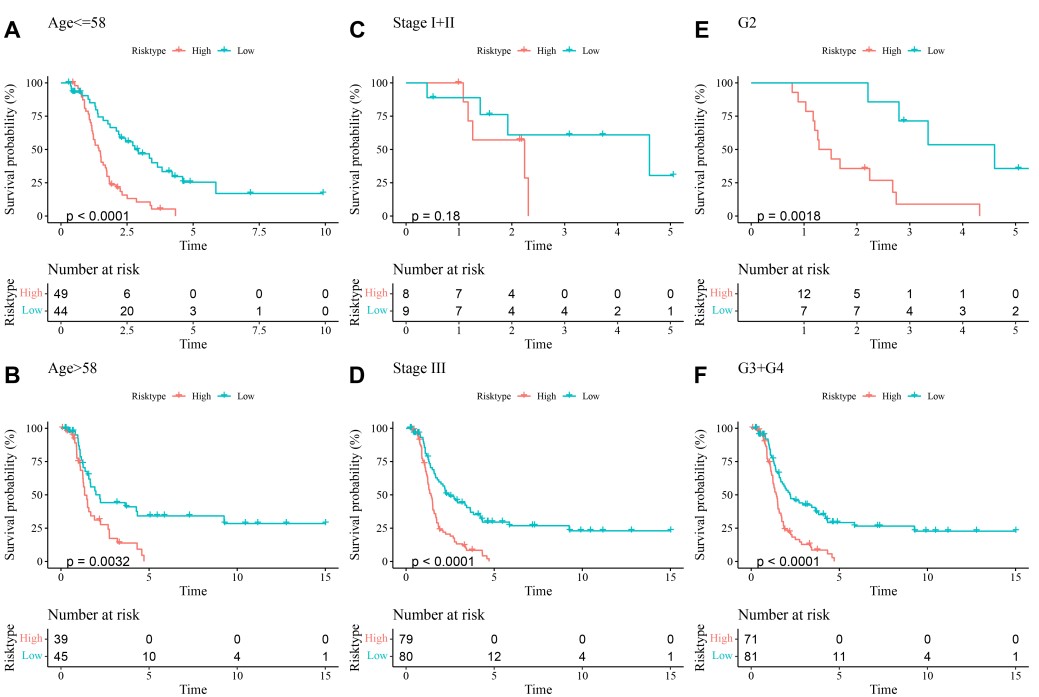

**Figure 5  DFI differences between high- and low-risk groups under different clinical characteristics.**
(A–B) DFI differences between the two risk groups under different Age; (C–D) DFI differences between the two risk groups under different Stage; (E–F) DFI differences between the two risk groups under different Grade.

importance to discover effective prognostic predictors and risk assessment model associated with relapse in OC. By performing scRNA-seq analysis, the present work observed that the proportion of Fibroblasts was reduced yet that of Epithelial cells was increased in relapse OC in comparison with primary OC. Subsequently, five independent prognostic genes related to the relapse in OC were identified to establish a RiskScore model, which was robust in predicting patients' prognosis. Compared to low-risk group, the patients in high-risk OC group showed worse outcomes under different clinical features and higher $IC_{50}$ values of the six predicted chemotherapy drugs. *In vitro* assays revealed that the invasive and migratory abilities of OC cells were inhibited by *KRT19* silencing. The present findings contributed to the prognosis assessment and individualized therapy of OC patients.

Since scRNA-seq analysis can provide novel understanding for the pathological mechanisms of OC progression and recurrence at the single-cell level (*Yuan et al., 2024*), this technique has been employed to help develop new signatures for OC to improve the current therapeutic paradigms (*Talukdar et al., 2021*). In this study, the single-cell atlas revealed four cell types (endothelial cells, epithelial cells, fibroblasts, myeloid cells), and the proportion of fibroblasts was markedly decreased yet that of Epithelial cells was notably increased in relapse OC samples in comparison to primary OC samples. Cancer-associated fibroblasts are implicated in the tumor growth and peritoneal metastasis of OC through remodeling extracellular matrix and tumor microenvironment (TME) (*Yang, Zhou &*

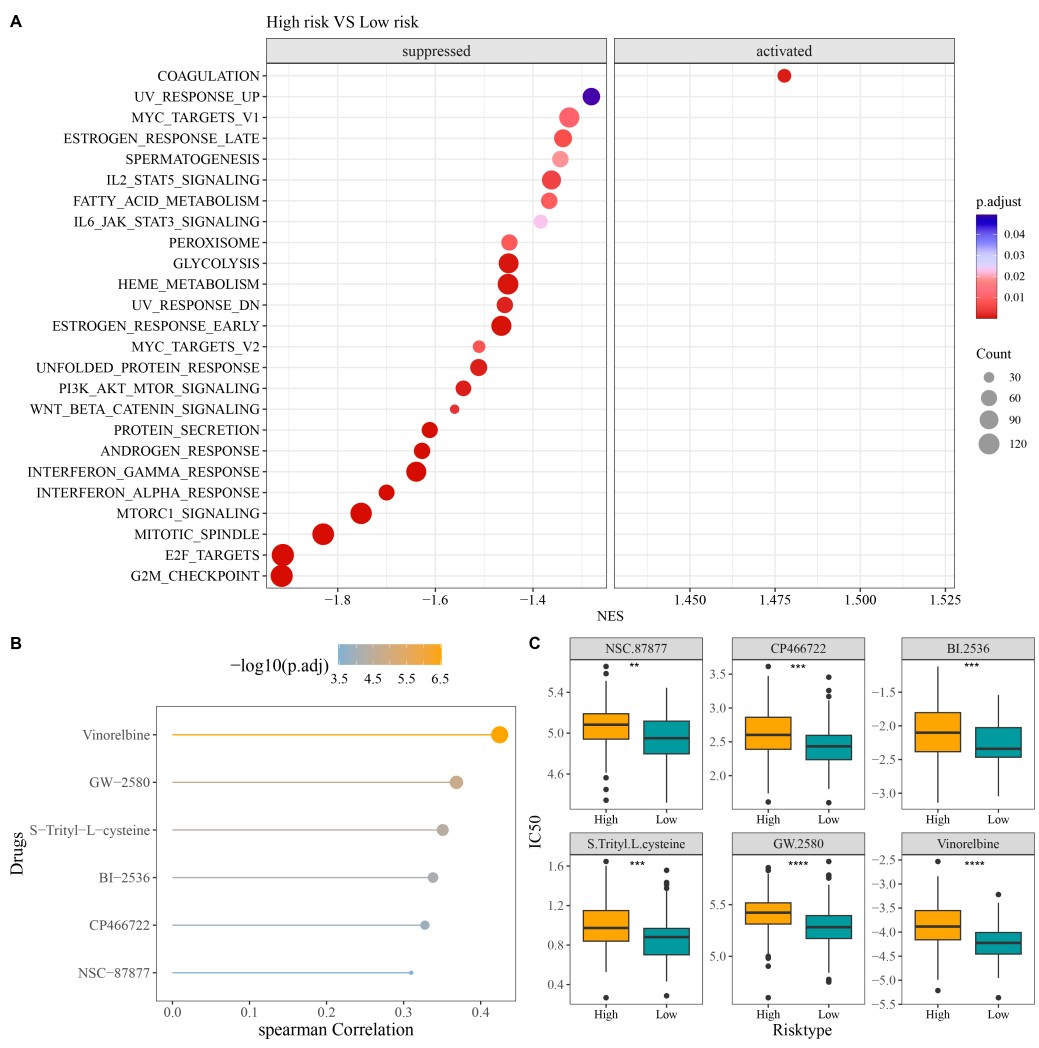

**Figure 6   Enrichment and drug sensitivity analysis.** (A) Difference of enriched pathways between high- and low-risk groups; (B) Spearman correlation analysis of RiskScore and IC$_{50}$ to screen the drugs with $p <$ 0.05 and |cor| > 0.3; (C) IC$_{50}$ values of different drugs in the two risk groups; **** denotes $p < 0.0001$; *** denotes $p < 0.001$; ** denotes $p < 0.01$.

*Huang, 2023*; *Ding et al., 2022a*). Moreover, OC derives from ovarian surface epithelium or serous intra-epithelial carcinoma (*Long et al., 2022*). Previous scRNA-seq study indicated that *MYBL2* in malignant epithelial cells is correlated with OC progression (*Shao et al., 2024*). epithelial-mesenchymal transition (EMT) functions crucially in OC metastasis and recurrence, during which epithelial cells reduce cell–cell adhesion and cell polarity while acquiring aggressive features to enhance mesenchymal phenotypes (*Padilla et al., 2019*). These findings demonstrated the underlying effects of epithelial cells in OC relapse.

Our relapse-related RiskScore model for OC consisted of two protective genes (*LDHA*, *NOP58*) and three risk genes (*NMU*, *KRT19*, *RPS23*), and exhibited strong robustness and independence in predicting the prognosis for patients with different clinical features. Compared to low-risk group, OC patients in high-risk group showed worse outcomes

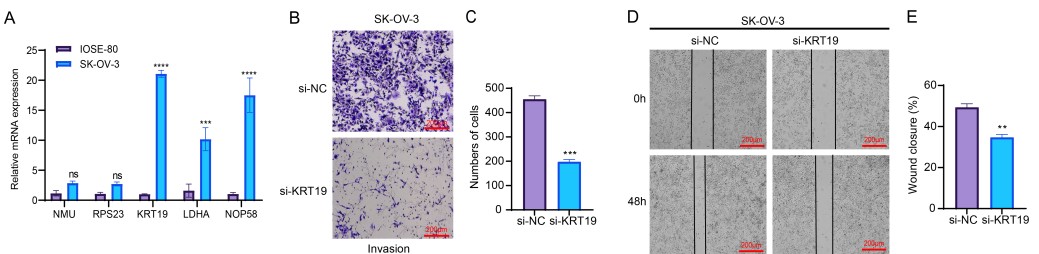

**Figure 7** **Effects of relapse-related prognostic genes on OC cells.** (A) Relative mRNA expressions of five prognostic genes in human normal ovarian epithelial cells IOSE-80 and OC cells SK-OV-3; (B–C) Transwell assay for the invasion ability of OC cells; (D–E) Wound healing assay for the migration capability of OC cells; ns indicates non-significant difference; **** indicates $p < 0.0001$; *** indicates $p < 0.001$; ** indicates $p < 0.01$.

with lower DFI and OS rate. Lactate dehydrogenase A (*LDHA*) as a critical enzyme in glycolytic pathway can modulate the synthesis and transport of lactate (*Dong et al., 2023*). *LDHA* is involved in the deteriorative progression of cancers *via* regulating multiple cellular mechanisms (*Wang et al., 2024b*). *Han et al. (2017)* manifested that miR-383 suppresses cell proliferation, invasion, and glycolysis in OC *via* targeting *LDHA*. It has been reported that *LDHA* is frequently overexpressed in OC than in normal tissue, which is often linked to poor survival outcomes and treatment resistance (*Xintaropoulou et al., 2018*). Nucleolar protein 58 (*NOP58*), a key component of box C/D small nucleolar ribonucleoprotein, functions importantly in maintaining cell homeostasis (*Wang et al., 2023*). *NOP58* is considered to be involved in tumor development and exhibits potential diagnostic and prognostic values in various cancers (*Qian et al., 2024*), including hepatocellular carcinoma (*Wang et al., 2021*), colorectal cancer (*Wu et al., 2020*), and prostate cancer (*Guo et al., 2024*). *NOP58* has also been identified as a relapse-associated hub gene in lung adenocarcinoma (*Shen et al., 2021*). However, studies of the role of *NOP58* in OC are limited. Neuromedin U (*NMU*) is a small neuropeptide with strong activity as an inducer of uterine smooth muscle contraction in rats, and plays a critical role in tumor genesis and metastasis of numerous cancers (*Przygodzka et al., 2019*). *NMU* and one of its receptors (*NMUR2*) are mainly expressed in the central nervous system such as hypothalamus as well as female ovarian and endometrial tissues (*Lin et al., 2015*). *NMU* signaling could promote endometrial cancer development, and the level of *NMU* is linked to the malignant grade and patient survival (*Lin et al., 2016*). Keratin 19 (*KRT19*), a member of the keratin family, encodes the cytoskeletal intermediate filament protein and its abnormal expression is vital in tumor progression (*Shi et al., 2024*). *KRT19* is recognized as a promising prognostic biomarker for OC, with high-expressed *KRT19* indicating worse prognosis and malignant progression (*Sun et al., 2023b*; *Ivansson et al., 2024*). Ribosomal protein S23 (*RPS23*) is a member of antimicrobial peptide with ancient origin and high conservation (*Ma et al., 2020*). *RPS23* plays an essential part in protein synthesis and immune response (*Wang et al., 2022a*). *RPS23* has been identified as a significant hub gene in hepatocellular carcinoma (*Sun et al., 2023a*), gastric cancer (*Dong et al., 2020*), and breast cancer (*Zhang et al., 2023b*). However, the specific regulatory mechanism of *RPS23* in cancers is less studied, especially in OC. Our

*in vitro* assays revealed that these five genes were all upregulated in OC cells and *KRT19* silencing notably inhibited the OC cell invasion and migration. Collectively, these genes might be involved in the initiation and development of OC, showing the potential to serve as predictors for OC relapse.

Furthermore, six chemotherapy drugs (Vinorelbine, GW-2580, S-Trityl-L-cysteine, BI-2536, CP466722, NSC-87877) were discovered to be notably correlated with RiskScore. Notably, high-risk group had higher $IC_{50}$ values, which inversely indicated that low-risk OC patients were more sensitive to the six drugs. In a Phase II clinical study, Vinorelbine is an anti-microtubule drug that exhibits similar efficacy to paclitaxel in treating OC (*Xu et al., 2021*). Vinorelbine combined with cisplatin could be employed as second- or higher-line palliative chemotherapy in advanced OC (*Yeon et al., 2023*). GW-2580 is a colony-stimulating factor 1 receptor kinase inhibitor (*Edwards et al., 2019*). *Moughon et al. (2015)* showed that administration of GW-2580 for advanced epithelial OC patients could reduce the infiltration of protumorigenic (M2) macrophage and markedly reduce the volume of malignant ascites. S-Trityl-L-cysteine, an inhibitor of human mitotic kinesin Eg5, is well recognized as an anticancer lead compound (*Radwan et al., 2019*). BI-2536 is an effective inhibitor of polo-like kinase 1, which is often overexpressed in OC and is a promising therapeutic target (*Rizvi et al., 2019*; *Wang et al., 2022b*). As an inhibitor of ataxia telangiectasia mutated (ATM) kinase, CP466722 suppresses the EMT and metastatic ability of drug-resistant lung cancer (*Shen et al., 2019*; *Jin et al., 2022*). NSC-87877 could inhibit the activation of Shp2 protein tyrosine phosphatase induced by epidermal growth factors in cell cultures (*Chen et al., 2006*). These findings have significant implications for guiding drug selection in the personalized treatment of OC.

Nevertheless, the present research also had several limitations. Firstly, a small sample size included in the single-cell dataset may limit a systematic identification of cellular heterogeneity and DEGs, potentially introducing bias. Subsequent studies should integrate more publicly available single-cell data or autonomously collect more primary and recurrent OC samples to establish a larger single-cell transcriptome cohort, so as to more accurately analyze changes in the TME and to enhance the robustness of the model-based data. Secondly, although this study verified the effects of *KRT19* on OC cell migration and invasion through *in vitro* experiments, we lacked in-depth functional validation for other key genes in the model, which limited a comprehensive understanding of its mechanism of action in OC. To provide a more reliable biological support for the model, we plan to examine the specific function of each candidate gene in OC recurrence and its regulatory mechanism based on *in vivo* functional experiments in the future. Finally, GSEA analysis revealed changes in the activity of several important pathways (*e.g.*, coagulation, Wnt/β-catenin, PI3K-AKT-mTOR, *etc.*) in high-risk OC group, but the specific roles or interactions of these pathways in the recurrence of OC remained unclear. For this reason, subsequent studies may combine immune microenvironment analysis, pathway intervention experiments, and spatial transcriptome technology to explore the synergistic roles and key nodes between the pathways, thereby further improving the understanding of the association between the model and the biological mechanisms of OC.

# CONCLUSION

In summary, this study established a relapse-related RiskScore model in OC with two protective genes (*LDHA*, *NOP58*) and three risk genes (*NMU*, *KRT19*, *RPS23*). The RiskScore showed a strong robustness and independence in predicting patient prognosis under different clinical features. Furthermore, six chemotherapy drugs (Vinorelbine, GW-2580, S-Trityl-L-cysteine, BI-2536, CP466722, NSC-87877) closely linked to the RiskScore were screened for the potential treatment of OC. Overall, our current findings may facilitate the drug selection, prognostic assessment, and personalized treatment of OC.

**Abbreviation**

| | |
|---|---|
| **ATM** | ataxia telangiectasia mutated |
| **AUC** | area under ROC curve |
| **BP** | biological process |
| **DEGs** | differentially expressed genes |
| **DFI** | disease-free interval |
| **EMT** | epithelial-mesenchymal transition |
| **FBS** | fetal bovine serum |
| **FDR** | false discovery rate |
| **GEO** | Gene Expression Omnibus |
| **GO** | Gene Ontology |
| **GSEA** | gene set enrichment analysis |
| **HR** | hazard ratio |
| **IC50** | half-maximal inhibitory concentration |
| **K-M** | Kaplan–Meier |
| **KRT19** | Keratin 19 |
| **LASSO** | least absolute shrinkage and selection operator |
| **LDHA** | lactate dehydrogenase A |
| **MSigDB** | Molecular Signature Database |
| **NMU** | neuromedin U |
| **NOP58** | nucleolar protein 58 |
| **OC** | ovarian cancer |
| **OS** | overall survival |
| **PCA** | principal component analysis |
| **P/S** | penicillin/streptomycin |
| **ROC** | receiver operating characteristic |
| **RPS23** | ribosomal protein S23 |
| **RT-qPCR** | quantitative real-time PCR |
| **scRNA-seq** | single-cell RNA-sequencing |
| **si** | small interfering |
| **ssGSEA** | single-sample GSEA |
| **TCGA** | the Cancer Genome Atlas |
| **TPM** | transcripts per million |
| **UMAP** | uniform manifold approximation and projection |
| **h** | hours |

## Funding

This project was supported by the Key Program of Bengbu Medical University (No. 2023byzd053) and Key Program of Natural Science Research of Higher Education of Anhui Province (No. 2024AH051291). The funders had no role in study design, data collection and analysis, decision to publish, or preparation of the manuscript.

## Grant Disclosures

The following grant information was disclosed by the authors:
Key Program of Bengbu Medical University: 2023byzd053.
Key Program of Natural Science Research of Higher Education of Anhui Province: 2024AH051291.

## Competing Interests

The authors declare there are no competing interests.

## Author Contributions

- Zhixin Jin conceived and designed the experiments, performed the experiments, analyzed the data, authored or reviewed drafts of the article, and approved the final draft.
- Xuegu Wang conceived and designed the experiments, analyzed the data, authored or reviewed drafts of the article, and approved the final draft.
- Xiang Li performed the experiments, prepared figures and/or tables, and approved the final draft.
- Shasha Yang performed the experiments, analyzed the data, prepared figures and/or tables, and approved the final draft.
- Biao Ding conceived and designed the experiments, prepared figures and/or tables, and approved the final draft.
- Jiaojiao Fei performed the experiments, prepared figures and/or tables, and approved the final draft.
- Xiaojing Wang conceived and designed the experiments, analyzed the data, authored or reviewed drafts of the article, and approved the final draft.
- Chengli Dou performed the experiments, analyzed the data, authored or reviewed drafts of the article, and approved the final draft.

## Data Availability

The public dataset used in this study is available at GSE63885 and GSE130000.
The raw data is available in GitHub.
- https://github.com/1ChengliDou/Raw-data.git
- 1ChengliDou. (2025). 1ChengliDou/Raw-data: Raw data (v.1.1.0). Zenodo. https://doi.org/10.5281/zenodo.15111394.

## Supplemental Information

Supplemental information for this article can be found online at http://dx.doi.org/10.7717/peerj.19764#supplemental-information.

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
