# Peer review of "Development of a relapse-related RiskScore model to predict the drug sensitivity and prognosis for patients with ovarian cancer"

_PeerJ, doi:10.7717/peerj.19764_

## Round 0.1 · original submission · Major Revisions

After careful consideration of the reviewers’ comments, I have decided that a major revision is necessary before the manuscript can be considered further. Please address all points raised by the reviewers, with particular attention to the major concerns, and provide a detailed point-by-point response.

Reviewer 1 ·

Basic reporting

1. The legend for Figure 7 should explain the meaning of "ns".
2. The scale bar in Figure 7b is not clear. Please make it clear and legible.
3. Why was KRT-19 chosen for the migration and invasion experiments?
4. Significance symbols should be added to Figure 1c for comparison. Only with significant differences can we draw conclusions about whether there is a decrease or increase.
5. The references are not up-to-date. Please try to cite references from the past three years.
6. The text needs to be polished, as there are some spelling errors, such as "with a 5-yrear overall survival (OS) rate," where "year" is misspelled.
7. Suggest using a structured abstract.

Experimental design

no comment

Validity of the findings

no comment

Reviewer 2 ·

Basic reporting

This study provides a novel RiskScore model for predicting ovarian cancer recurrence, guiding personalized treatment and revealing molecular mechanisms, thus advancing precision medicine and improving patient outcomes.
#1. Line 25-27, What is the significance or purpose of performing single-cell analysis, How does it help in constructing the risk model. Are there specific cell subtypes or signaling pathways that critically contribute to the process of recurrence?
#2. Line 68-73, In describing the findings of the scRNA-seq analysis, it is valuable to further explore the functional dynamics and intercellular interactions of distinct cell types during tumor recurrence. For instance, do specific cell subpopulations or signaling pathways critically contribute to the recurrent process?
#3. Line 74-82, The conclusion should clearly highlight the key findings of this study and their significance for clinical practice, while also outlining potential future research directions and challenges. For instance, strategies for integrating the RiskScore model into current clinical management pathways should be discussed, along with anticipated obstacles and feasible solutions for real-world implementation.
#4. Line 192, “Five independent prognostic genes related to relapse in OC were identified”. The title should not use passive sentences. Check all the titles.
#5. Line 262-269, the differences between the single-cell profiles of this study and those of other single-cell profiles of ovarian cancer lack comparison.
#6. The Wnt/β-catenin and PI3K-AKT-mTOR signaling pathways often exhibit interactive regulation in cancer; however, they are presented separately in the manuscript without analysis of their potential synergistic effects or functional complementarity. It is recommended to include a discussion on the possible cross-talk between these pathways and their combined mechanisms in the progression of ovarian cancer (OC).
#7. For GSEA analysis, The study notes coagulation pathway activation and suppressed cell cycle and immune pathways in the high-risk group, but does not explore their potential mechanisms in ovarian cancer recurrence. For example, is coagulation activation linked to tumor microenvironment remodeling, immune escape, or enhanced metastatic potential? A deeper mechanism discussion integrating existing literature is needed.
#8. While evidence indicates that the RiskScore model holds promise for facilitating personalized therapy, the practical challenges of its clinical implementation remain insufficiently explored. It is essential to thoroughly examine the potential obstacles to integrating this model into current clinical pathways, along with strategies to overcome them. Key considerations include cost-effectiveness analysis, the willingness of clinicians and patients to adopt the model, and the technical feasibility of its integration into routine practice.
#9. What is the specific clinical application of RiskScore and how does it improve patient outcomes.
#10. Please deepen the limitations of this article.

Experimental design

no comment

Validity of the findings

no comment

---

## Round 0.2 · accepted · Accept

Both reviewers have evaluated your revised version and found that you have adequately addressed their previous concerns. The manuscript has been substantially improved and now can be considered for publication.

Reviewer 1 ·

Basic reporting

no comment

Experimental design

no comment

Validity of the findings

no comment

Additional comments

In this study, they developed a recurrence-based ovarian cancer (OC) risk score model, combined with single-cell sequencing and batch RNA data, successfully screened five independent prognostic genes (LDHA, NOP58, NMU, KRT19, RPS23), and verified their prognostic predictive ability and drug sensitivity association. The study design was rigorous, the data analysis was comprehensive, and the conclusions had the potential for clinical translation and met the publication criteria.

Reviewer 2 ·

Basic reporting

no comment

Experimental design

no comment

Validity of the findings

no comment

Additional comments

In this study, by integrating single-cell sequencing and bulk RNA-seq data, we constructed and validated a 5-gene recurrence risk score model for ovarian cancer (OC) with high potential for clinical translation. The research design is rigorous, the data analysis is comprehensive, and the conclusion is innovative, especially in the use of single cell technology to analyze the mechanism of recurrence.